# Dynamic Segmentation of Sensor Events for Real-Time Human Activity Recognition in a Smart Home Context

**DOI:** 10.3390/s22145458

**Published:** 2022-07-21

**Authors:** Houda Najeh, Christophe Lohr, Benoit Leduc

**Affiliations:** 1IMT Atlantique, Lab-STICC, 29238 Brest, France; christophe.lohr@imt-atlantique.fr; 2Delta Dore Company, 35270 Bonnemain, France; bleduc@deltadore.com

**Keywords:** real-time human activity recognition, dynamic segmentation, smart building, event correlation, temporal correlation, triggering sensor

## Abstract

Human activity recognition (HAR) is fundamental to many services in smart buildings. However, providing sufficiently robust activity recognition systems that could be confidently deployed in an ordinary real environment remains a major challenge. Much of the research done in this area has mainly focused on recognition through pre-segmented sensor data. In this paper, real-time human activity recognition based on streaming sensors is investigated. The proposed methodology incorporates dynamic event windowing based on spatio-temporal correlation and the knowledge of activity trigger sensor to recognize activities and record new events. The objective is to determine whether the last event that just happened belongs to the current activity, or if it is the sign of the start of a new activity. For this, we consider the correlation between sensors in view of what can be seen in the history of past events. The proposed algorithm contains three steps: verification of sensor correlation (SC), verification of temporal correlation (TC), and determination of the activity triggering the sensor. The proposed approach is applied to a real case study: the “Aruba” dataset from the CASAS database. F1 score is used to assess the quality of the segmentation. The results show that the proposed approach segments several activities (sleeping, bed to toilet, meal preparation, eating, housekeeping, working, entering home, and leaving home) with an F1 score of 0.63–0.99.

## 1. Introduction

Environmental regulations do not consider the building as a simple consumer of energy. Instead, they put the human being at the center of concerns, so that the building is perfectly integrated into its ecosystem [1]. The buildings of tomorrow must be fully in line with a vision of a renewed society, where comfort is in harmony with the dynamic management of energy. To reach these goals, it is also important to develop new control functions that offer stronger interactions with the residents. A number of important studies [2,3] have shown that human behavior is the origin of significant differences in energy consumption (more than 20%). From this perspective, the building is seen as an ecosystem in which the equipment must meet the needs of the occupants internally, externally, and in the surrounding neighborhood. The building ecosystem is consequently complex, insofar as it is made up of heterogeneous coordinated systems for optimized management.

Each building is characterized not only by its location and configuration, but also by its occupants. The needs of the occupants change over time, and the control system must be able to adjust. The arrival of digital technology in buildings (such as building information modeling (BIM), the Internet of things (IoT), and sensors) provides additional information that can be used for building energy management [4]. This is possible thanks to targeted and adapted processing that uses the data generated by these communicating objects to understand occupants’ behavior, evaluates various equipment performances, anticipates faults, and automates home services. Feedback to residents themselves can also been considered as an added value (schedule of presence/absence, energy-consuming behavior). In fact, besides the techniques that interact with the building directly, many researchers focus on occupants to improve the energy efficiency in smart homes. Their goal is to enhance occupants’ knowledge of their activities and their impacts on energy consumption. Then, their practices can be adjusted to reduce wasted energy and improve their comfort. Therefore, many recommendation methods and forms of energy feedback have been proposed for occupants [5]. These tools aim to provide insights on the impacts of their activities on building energy consumption. This information helps residents handle and understand the finances related to energy usage. On the other hand, the recommendation systems could support occupants in evaluating their activities regarding energy consumption and comfort. Their past activities could be analyzed to suggest better practices [6].

It is from this perspective that the Delta Dore company intends to pursue its innovation efforts. Developing new solutions for the recognition of human activities is a question of offering services regarding energy consumption, health, and comfort in a home [7].

With the massive arrival of sensors that can communicate at low cost, the building sector is currently experiencing an unprecedented revolution: the building is becoming intelligent, that is, it offers new services to occupants related to security, energy management, and comfort. Article 23 of the 2012 thermal regulations (France) imposes the minimum measurement of certain consumption items, which promotes the deployment of sensor networks in new buildings. In addition, several research projects show the interest of public bodies and companies (e.g., Delta Dore) in guaranteeing overall performance (actual total consumption, interior comfort) after rehabilitation using these sensor networks. In addition to the various aspects of comfort and energy consumption, these sensors also make it possible to estimate the behaviors of the occupants, which determine energy consumption, by estimating the number of occupants per zone and their metabolic contribution, activities, and routines.

The recognition of human activities is fundamental to many services [8,9], but providing sufficiently robust HAR systems that could be deployed in an ordinary real environment remains a major challenge. The existing works in the literature have focused mainly on recognition through pre-segmented sensor data (i.e., dividing the data stream into segments, each defined by its beginning and its end) [10]. In fact, in a smart building, sensors record the actions and interactions with the residents’ environment over time. These recordings are logs of events that capture the actions and activities of daily life. Most sensors only send their status in the case of status change, not only to reserve the power of battery, but also to not overload wireless communications. Moreover, sensors may have different triggering times. This results in irregular and scattered time series sampling. Therefore, the recognition of human activities in a smart home is a pattern recognition problem in a time series with irregular sampling. Segmentation techniques provide a representation of sensor data for human activity recognition algorithms. A time window method is commonly used in events’ segmentation for HAR. However, in the context of a smart home, sensor data is often generated episodically over the time, so a fixed size time window is not feasible. One of the challenges of dynamic segmentation based on data streaming is how to correctly identify whether two sensor events with a certain temporal separation belong to the same activity or not.

To perform real-time recognition, the objective is to determine if the last event that has just occurred belongs to the current activity, or if it is the sign of the start of a new activity. For this, we consider the correlation between sensors in view of what can be seen in the history of past events. The proposed algorithm contains three steps: (i) verification of sensor correlation (SC), (ii) verification of temporal correlation (TC), and (iii) identification of the activity triggering the sensor.

This paper is organized as follows: Section 2 presents the problem statement, as well as the objective and the main contribution of the paper. The related works are summarized in Section 3. Section 4 is devoted to a description of the proposed approach. To assess the efficiency of the proposed methodology, simulation results for a real case study are presented in Section 5. Section 6 discusses the findings of the literature review, the proposed methodologies, their limits, and the results. Finally, concluding remarks are given in Section 7.

## 2. Problem Statement and Research Objective

In an instrumented building, sensors may trigger at different times [11]. This results in irregularity in the data sampling in the time series. Therefore, HAR in a smart building is a pattern recognition problem in a time series with irregular sampling [12].

In this section, the existing methodologies for sensor data stream segmentation in a smart building are firstly highlighted. These approaches provide a representation of sensor data for HAR algorithms. Then, the challenges regarding the temporal complexity of human activity data in real use cases are discussed in details.

### 2.1. Existing Works

In an HAR task, each piece of sensor data needs to be divided into chunks, in a windowed manner. Some windowing approaches [13,14,15,16,17] are more appropriate for real-time activity recognition than others. Real-time is a necessity because a smart building should be reactive. For example, let us take the example of a health service in an smart building. It would be useless to detect the fall of an elderly person one hour after it has happened; however, it would be useful to detect it at the moment of the fall. Another example is that of lowering energy use. Indeed, it is useless to decide to cut the heating or lower the shutters only several hours after having registered the activity of the inhabitants.

This section details the most used data segmentation techniques in a real-time HAR context.

#### 2.1.1. Time Window

Figure 1 illustrates an example of a sequence of activities, represented by Activity 1, Activity 2, Activity 3, and Activity 4, along with the corresponding events in temporal order. On the other hand, the interval of time between each pair of events is variable.

Time windowing (TW) techniques consist of dividing the data stream into time segments with a regular time interval, such as every 60 s, as described in Figure 2.

The selection of the optimal duration of the time interval is the biggest challenge for these techniques. In fact, if the time interval is very large, the events could be common to many activities, and consequently the predominant activity in the time window will have the more influence in the label’s choice [18]. TW techniques are commonly used in the segmentation of events for real-time activity recognition. This technique is more favorable to the sensor time series with regular or continuous time sampling. However, in the smart home context, sensor data are often generated in a discrete form along the timeline, where a fixed-size time window is not suitable.

#### 2.1.2. Sensor Events Window

A sensor events windowing technique consists of dividing the whole sequence into a set of sliding windows with an equal number of sensor events [13], for example, every 20 sensor events, as demonstrated in Figure 3.

Each window is labeled with the last event’s label in the window, and the sensor events that precede the last event in the window define the last event’s context.

This is easier to implement; however, the challenge consists of the fact that the actual occurrence of the activities is not intuitively reflected [15]. In fact, one sliding window could cover two or more activities, or  events belonging to one activity can be spread into several sliding windows. Furthermore, if several activities occur simultaneously, this technique is unable to segment events. In addition, this window type differs in terms of duration, and it is consequently impossible to interpret the time between events.

#### 2.1.3. Activity-Based Event Window

To solve these issues regarding the sensor event windowing technique, the activity-based event windowing method segments the events into blocks that are compatible with the occurrence of each activity, as presented in Figure 4. This approach precisely defines the boundaries of the activity, but it may take a longer time until adequate information to define one segmentation is received. This is a challenge. Moreover, the challenge is how to determine whether two sequential events belong to identical activities or not. In the following, we discuss a two-step dynamic segmentation approach.

### 2.2. Objective

In the literature, an important number of research works have been developed in the field of human activity recognition. However, a number of problematic challenges remain outstanding, such as human activity recognition in real time. The main contribution of this work is to develop a real-time HAR based on dynamic event segmentation and the trigger sensor’s activity.

Dynamic windowing. Most existing solutions dealing with HAR are offline and aim to recognize activities based on pre-segmented datasets, while real-time HAR based on streaming ambient sensor data residue is problematic. This research work discusses a novel dynamic online events segmentation approach, which facilitates real-time activity recognition. Moreover, with this approach, we could predict the current activity label in a timely manner. The objective of dynamic segmentation makes it possible to delimit the beginnings and ends of each activity.Trigger sensor. This paper considers the importance of knowing the triggering sensor of an activity for the HAR in the context of daily living. Once the beginnings and endings of each activity are determined by the real-time dynamic segmentation algorithm, knowledge of the triggering sensor helps identify the activity.

The identification of the triggering sensors of an activity consists of selecting the one that appears first most often among the occurrences of this activity in the dataset.

### 2.3. Contribution

In summary, our contribution demonstrates that, by the dynamic sensor windowing and the knowledge of the triggering sensor, we can improve the classification of activities.

## 3. Real-Time Recognition of Human Activities in Smart Homes

The HAR task proceeds from pattern recognition, where the techniques are decomposed into two categories.

### 3.1. Ontology-Based Approaches

Various techniques have been proposed in the literature to tackle the HAR task, and most of the basic ones focus on the location of sensors in the house because it is a crucial part of the context. They assume that the semantics of a spatial position are static; that is why they focus on how to specify the semantics of a spatial position. They also focus on how to identify the spatial position of occupants [19].

Location-based techniques [20,21] work well to some extent. However, they are unable to recognize human activities with enough accuracy. For example, if the occupant stays in a kitchen, she or he may be cooking food or putting the dishes away; the efficiency of this method is limited because a location hosts various activities. For now, research has failed to handle these changes in semantics.

Another type of ontology-based approach is the thing-based technique [22], which identifies, in a dynamic manner, the change in activities by using remote sensors to identify the objects that the occupant is interacting with [23]. For example, to recognize an occupant activity, [24] developed a system of semantics identification that identified the activity space from objects forming the immediate environment of the occupant.

Furthermore, these techniques try to overcome the diversity of activities in the most comprehensive way possible. However, they require a complete knowledge of the domain to elaborate activities models. They are not robust to handle settings change and uncertainty, and they have implementation problems regarding the recognition accuracy of the used sensors depending on their cost. In addition, they propose following expert knowledge to model activities, which is time consuming and difficult to maintain in the case of environment evolution.

### 3.2. Data-Driven Approaches

Another technique to recognize human activities in real time is to monitor the appliances’s status in order to understand the behavior of the different loads in a smart home area.

In the scientific literature, several load monitoring techniques can be implemented. They can be divided into two categories [25], as follows:Intrusive load monitoring (ILM) is a data collection methodology where each appliance node is equipped with many devices to detect the sensor events and therefore characterize in detail the occupant activities using deep learning techniques, for example.The databases generated by these systems can be labeled manually (i.e., the user labels the monitored appliance) or automatically (i.e., the system is trained with examples from distinctive appliances and then recognizes the appliance that is being used). Generally, manual setup ILM systems outperform automatic setup ILM systems. Accuracy of the results is the main benefit of this method, but it requires expensive and complex installation systems.Non-intrusive load monitoring technique (NILM) is an alternative operation, in which one single monitoring device is installed at the main distribution board in the home area. An algorithm is performed to determine the operation’s stat for each appliance. The main advantage of the NILM approach is the fact that only one single monitoring device is needed. Therefore, it lowers the cost significantly at the home level. The main disadvantage is the lower accuracy compared to ILM systems (in particular, those with manual labeling [26]).

In general, the appliances of a household can be categorized in the following classes: (i) finite-state appliances such as dishwashers or fridges, which have different states, each one having its duration (cyclic or fixed) and its own power demand; (ii) continuously varying appliances, such as computers, which have different states and behavior that is not periodic; (iii) on/off appliances, such as light bulbs with a fixed demand of power; (iv) permanent demand appliances, such as alarm-clocks, which are always in “ON” mode with a fixed power request.

The appliances can be recognized by “event-based” approaches that detect the ON/OFF changes, or by “non-event-based” or “energy-based” approaches that detect whether an appliance is ON during a sampled duration.

Different sensor measurements and feature data are required for these techniques, such as voltage and current. The most important parameter in the complexity level of the discretion methods is the sampling rate. It affects not only the type of feature that can be measured, but also the type of algorithm that can be used. A detailed review of the features and algorithms is included in [27].

Another important issue in data-driven approaches is related to the quality of data. In this work, we are interested in dealing with ambiguous outliers’ detection in both training and testing data and in the insufficiency of labeled data. Outliers could be detected using a multiclass deep auto-encoding Gaussian mixture model, for example [28]. This is a set of individual unsupervised Gaussian mixture models that helps the deep auto-encoding model to detect ambiguous outliers in both training and testing data.

Although well-documented, most HAR techniques only use pre-segmented datasets. However, human activities need to be monitored in real time in many scenes. This requires the HAR algorithm to steer clear of pre-segmented sensor data and focus on streaming data.

## 4. Proposed Methodology for Human Activity Recognition

### 4.1. Key Ideas

This work tackles the challenge of how to determine whether two sequential events belong to identical activities or not. It focuses on a two-step dynamic segmentation approach that incorporates the computation of sensor correlation and temporal correlation.

In the following, we describe the proposed methodology for real-time human activity recognition in smart homes, designed as a classifier of a semantic time series. It relies on a real-time dynamic windowing of sensor events and trigger sensor identification for each activity combined as depicted in Figure 5. To summarize, this algorithm processes real-time streaming sensor data, which comprises two phases. In the offline phase, the dataset is cleaned, and the data are sampled. The online phase is a three-step methodology: the checking of event correlation, the checking of time correlation, and the identification of the triggering sensor.

For a better understanding of this technique, we will follow the step-by-step workflow described in this schema.

### 4.2. Step 1: Dynamic Windowing in Real Time

#### 4.2.1. Computation of Event Correlation

In a general case, an event sequence can be represented as $E=\{e_{1},e_{2},\ldots ,e_{n}\}$, where $e_{i}$ refers to the *i*th event, and each $e_{i}\in E$ contains a vector of information $<T_{i},s_{i},V_{i}>$ where $T_{i}$, $s_{i}$, and $V_{i}$ represent respectively:$e_{i}$: *i*th event.$T_{i}$: time stamp of the *i*th event (date, hour:minute:second).$s_{i}$: sensor name of the *i*th event.$v_{i}$: sensor value of the *i*th event.

Thus, one of the challenges is how to split the sensor sequence into a series of event chunks that are associated with corresponding activities.

In this paper, the dynamic segmentation of events is proposed. It is determined using the Pearson product moment correlation (PMC) coefficient between the events [29]. PMC, or more formally $\rho_{X,Y}$, is a measure of the linear correlation between two variables, X and Y (two events in our case study). It is calculated using Equation (Equation 1).
(1)$$\rho_{X,Y}=\frac{cov(X,Y)}{\sigma_{X}\sigma_{Y}}$$
where:cov(X,Y) is the covariance of *X* and *Y*.$\sigma_{X}$ and $\sigma_{Y}$ are the standard deviation of *X* and *Y*, respectively.

It produces a value between −1 and 1. Three cases are distinguished:*X* and *Y* are totally positively correlated if $\rho_{X,Y}=1$.*X* and *Y* are totally negative correlated if $\rho_{X,Y}=-1$.*X* and *Y* are not correlated if $\rho_{X,Y}=0$.

In the case of the smart home test bed used in this work and presented in Section 5.1, each sensor is encoded with an identifier of the location where the sensor is installed.

Let us consider the following example: Figure 6 shows an example of the PMC sensor correlation rate between nine sensors installed in a home configuration. As is depicted, the sensors 3, 4, 5, 6, and 7 are highly correlated.

Thus, the PMC value (i.e., $\rho_{X,Y}$) between two geographically close sensors is more likely to be higher than two sensors geographically far from each other. It suggests also that sensors in the same or adjacent regions are more likely to be triggered together or sequentially.

Thus, the threshold value for $\rho_{X,Y}$ is set at 1 (i.e., the sensors are highly correlated). Therefore, given the incoming event sequence $E=\{e_{1},e_{2},\ldots ,e_{n}\}$, if the $\rho_{X,Y}$ between event $e_{i}$ (triggered by sensor $s_{i}$) and event $e_{i+1}$ (triggered by sensor $s_{i+1}$) is 1, it can be considered that event $e_{i}$ and event $e_{i+1}$ are correlated.

#### 4.2.2. Computation of Temporal Correlation of Sensor Events

The most known methods used in sensor event segmentation for real-time HAR are time windowing techniques [29,30,31]. However, in the smart home context, sensor data are often generated in a discrete manner, so a fixed time window is not practical along the schedule.

A segmentation is a set of segments $S=\{s_{1}\ldots s_{m}\}$. Each segment contains a set of sensor events, triggered by a human during an activity. A segment has a label, which is the label of the activity. It has a start date and an end date. It is encoded in the template of

$s_{j}=<e_{first}\ldots e_{last};label_{j}>$, so $e_{first}=<T_{first},s_{first},V_{first}>$ and $e_{last}=<T_{last},$$s_{last},V_{last}>$. Then $s_{j}^{first}=T_{first}$ and $s_{j}^{last}=T_{last}$.

An activity has several occurrences in the database, so an activity can be represented by several segments.

Determining whether two events with a certain temporal separation belong to the same activity or not is the main challenge of dynamic segmentation.

Let us consider two events with a high dependency between sensors and a large time interval. These two events should not be in the same sliding window. Consequently, a measure based on time correlation is employed to decide if two events are temporally dependent or not. In this work, a time-correlation-based method is investigated in order to identify whether two events are time correlated or not.

For an existing segmentation, the first and last time stamps are designated as $T_{first}$ and $T_{last}$, respectively. Then, each incoming event $e_{i}\in E$ is manipulated twice with $T_{first}$ and $T_{last}$, utilizing Equations (Equation 2) and (Equation 3), respectively.
(2)$$T_{cor}^{first}=\frac{\mathrm{time}\;\mathrm{distance}}{\mathrm{Maximum}\;\mathrm{Time}\;\mathrm{Span}}=\frac{T_{last}-T_{first}}{T_{max}}$$
(3)$$T_{cor}^{last}=\frac{\mathrm{time}\;\mathrm{distance}}{\mathrm{Maximum}\;\mathrm{Time}\;\mathrm{Interval}}=\frac{T_{last}-T_{first}}{P}$$
where:$T_{max}$ is the maximum duration for a given activity.*P* is the probability of occurrence of an activity in a given area. It is a constant value that represents the probability of occurrence of activity in a zone. For example, “meal preparation” is supposed to occur in the kitchen, with a probability equal to 100%. In a dataset, the sequence of events is recorded in a chronological order, so $T_{last}-T_{first}>0$ and $T_{max}>0$. With regard to the threshold *P*, it is related to the duration of activities.

However, the disadvantage of this approach is that there may be significant differences between the durations of various activities. For example, on analysis of the longer term dataset, the average duration of sleeping is 3 h: 35 min: 57:45 s, while the average duration of eating is 09 min: 55:23 s. To address this issue, different threshold parameters are set corresponding to each functioning area (Table 1).

In summary, the following Algorithm 1 describes the pseudocode of the algorithm dealing with real-time streaming events. It is a two-step methodology: (i) the computation of sensor correlation (SC); (ii) the computation of temporal correlation (TC).

The sensor correlation shows that sensors in the same zone or close functioning areas are more likely to be triggered together or sequentially. The temporal proximity of the events corresponds to a geographical proximity of the sensors, which justifies the relevance of the approach.

For an incoming event $e_{i}$, both sensor and temporal correlations will be conducted with all the existing segmentations. Then, if SC and TC are verified, $e_{i}$ will be added to the segmentation that aligns with both SC and TC results. However, if none of the existing segmentations can be identified, a new segmentation can be initialized with $e_{i}$.
**Algorithm 1** Real-time dynamic windowing**Require:** Sequence of events $E=\{e_{1},e_{2},\ldots ,e_{n}\}$**Ensure:** set of segmentations $S=\{s_{1},s_{2},\ldots ,s_{n}\}$  **if**
*S* not exist **then**  Initialize $s_{1}$ and add $e_{1}$ to $s_{1}$**else if then**S exist   **for**
$s_{j}$: j = 1 to n **do** <>    **if**
$s_{j}\neq ⌀$
**then**     SC = sensor correlation ($e_{j}$, $e_{last}$)     $T_{cor}^{first}$ = time correlation ($e_{j}$, $e_{first}$)     $T_{cor}^{last}$ = time correlation ($e_{j}$, $e_{last}$)     **if** SC = True AND ($T_{cor}^{first}$ = True AND $T_{cor}^{last}$ = True) **then** add $e_{j}$ to $s_{j}$     **else if** SC = True AND ($T_{cor}^{first}$ = False OR $T_{cor}^{last}$ = False) **then** clear $s_{j}$ and start a new segmentation from the next event     **end if**    **else if**
$s_{j}=⌀$
**then** Initialize $s_{j+1}$ and add $e_{j}$ to $s_{j+1}$    **end if**  **end for****end if**

### 4.3. Step 2: Identification of Triggering Sensors

The calculation of the correlation between sensors seeks to recover the sensors that are correlated. Highly correlated sensors (i.e., whose PMC value is equal to 1) are frequently co-located in the same area (i.e., there is a geographical proximity between the sensors).

Among many co-located sensors, some of them are the triggering sensors for each activity. The identification of the triggering sensors of an activity consists of first selecting the one that appears most often among the occurrences of this activity in the dataset. In this work, this criterion is used to recognize the activity once the beginning and the ending of a segment are determined.

### 4.4. Algorithm Performance Evaluation

Validation is an essential part of evaluating the algorithm performance. Here, to validate the proposed algorithm, we verify whether it can accurately estimate occupant activities.

In statistical analysis, the F-score is an indicator about the accuracy of a defined test. It is calculated from the precision (P) and recall (R) of the test defined by Equations (Equation 4) and (Equation 5), where:TP: represents the true positive.FP: represents the false positive.FN: represents the false negative.

The F1 score is the harmonic mean of the precision and recall. An F-score equal to 1 indicates a perfect precision and recall, and an F-score equal to 0 indicates that either the recall or the precision is null.
(4)$$P=\frac{TP}{TP+FP}$$
(5)$$R=\frac{TP}{TP+FN}$$
(6)$$F-\mathrm{score}=2\times \frac{P\times R}{P+R}$$

An example of F-score calculation of sleeping activity is given in Figure 7.

## 5. Case Study

To test the methodology and its generalization, it was implemented in a real case study. In this section, to be consistent with previous sections, eight activities are used to illustrate the methodology. A two-year dataset from November 2010 to June 2011 is used for testing.

### 5.1. Data Set Description

The experiment was conducted on the CASAS dataset [32], specifically Aruba, as introduced by Washington State University. This dataset contains sensor data that was collected in the home of a volunteer adult. The resident in the home was a woman. The woman’s children and grandchildren visited on a regular basis.

The following activities are annotated within the dataset. The number in parentheses is the number of times the activity appears in the data.

Meal preparation (1606);Relax (2910);Eating (257);Work (171);Sleeping (401);Wash dishes (65);Bed to toilet (157);Enter home (431);Leave home (431);Housekeeping (33).

The events are generated from motion sensors (these sensor IDs begin with “M”), door closure sensors (these sensor IDs begin with “D”), and temperature sensors (these sensor IDs begin with “T”). The layout of the sensors in the home is shown in Figure 8.

Figure 9 represents an extract of the dataset with raw sensor data, where the annotated sensor events in the dataset include different categories of predefined activities of daily life, while the untagged sensor events are all labeled with “Other Activity”.

For reasons of software development, the motion sensors and door sensors with “SensorValue” of “ON/OPEN” and “OFF/CLOSE” are mapped to “1” and “0”, respectively. “Date” is converted from the form of “yyyy-mm-dd H:M:S” to epoch time in milliseconds relative to the zero hour of the current day.

### 5.2. Implementation and Results

In this section, we will firstly discretize the sensor data at a fixed rate. Secondly, we will apply the real-time dynamic segmentation of events to determine the beginning and the ending of each segment and then identify the activity using the trigger sensor. Finally, we will compare the simulated and real activities using the F-score indicator.

### 5.3. Discretization of the Data Set

In this work, the data are sampled with a sampling rate equal to 1 s. This value is justified by the fact that, if we increase the sampling rate, we lose information. Therefore, the smaller the value is, the richer the segment is in events, and the more accurate the segmentation will be. Figure 10 and Figure 11 show, respectively, the data evolution of three motion detection sensors (M003, M005, and M007), as well as the temperature sensors.

### 5.4. Trigger Sensor Identification for Each Sensor

Figure 12, Figure 13 and Figure 14 show histograms of the first sensor of an activity for occurrences of the activity in the dataset.

For some activities, such as sleeping, bed to toilet, work, and eating, the triggering sensor is always the same (M003, M004, M026, and M014, respectively). However, for other types of activities, such as housekeeping, different sensors can trigger the activity (M008, M013, M020, and M031).

### 5.5. Results of Activities’ Recognition

Figure 15 and Figure 16 show, respectively, the results of the real-time dynamic sensor event segmentation algorithm (i.e., the identification of the beginning and end of activities) for five activities—sleeping, bed to toilet, meal preparation, leave home, and enter home—over 24 h on 4 November 2010.

Over 24 h on 4 November 2010, the activity sleeping happened two times. Table 2 shows the segments of the two real and simulated sleeping activities (i.e., the beginning and the end of each activity).

Table 3 shows the evaluation of the quality of segmentation for the sleeping activity using F-score.

Over 24 h on 4 November 2010, the activity eating happened four times. Table 4 shows the segments of the different eating activities.

Table 5 shows the evaluation of the quality of segmentation for the eating activity using F-score.

In this paper, only the details about sleeping and eating activities are given. The details for other activities are omitted. Table 6 summarizes the segmentation performance (precision, recall, and F-score) on 4 November 2010 for all the activities.

Table 7 summarizes the segmentation performance (precision, recall, and F-score) on 5 November 2010 for all the activities.

The problem of concurrent or multi-user activities recognition associated with correlated sensors’ events remains a critical research challenge. For example, the activity of housekeeping in the kitchen and meal preparation are both associated with a certain number of sensors, such as common motion detection sensors for example. If such activities occur simultaneously or are intersecting, this approach may not be achievable in segmenting such sensor events. Even if occurring in both of the datasets in this work, such situations are infrequently considered. However, scenarios where there is more that one resident are quite common and must be given much attention by the research community.

### 5.6. Comparison with Other Methods

The proposed method is a segmentation method to be used by recognition methods, not a recognition method in itself. In the literature, all the existing segmentation methods are offline, and there is not much attention to real-time segmentation.

By identifying the triggering sensor, it turns out that the proposed segmentation technique makes it possible to recognize the activity, but that is a side effect. Therefore, in this work, the results are compared to those of the data labeled in the dataset.

Despite certain biases in the dataset and the topology of the house, our performance is less good than the algorithms of the domain, but it is not bad either. As a comparison, the recognition performance (i.e., the average F-score) for eight activities (sleeping, bed to toilet, eating, housekeeping, relax, work, enter home, and leave home) of the proposed algorithm is 0.766, while this value is equal to 0.73 using the decision tree technique presented in [33], 0.765 using sensor-event-based windowing technique [13], 0.760 using a combination of SWMI (fixed-length sensor windows with mutual-information-based weighting of sensor events) and SWTW (fixed-length sensor windows with time-based weighting of the sensor events) techniques [13], 0.773 using SWTW technique [13], and 0.83 using the naive Bayes tree technique presented in [33].

## 6. Discussion

To recognize an activity in real time, two steps are followed:Step 1: Calculation of spatio-temporal correlationGiven an incoming event, the sensor correlation and time correlation will be performed with all existing segmentations. Subsequently, it will be added to the segmentation that aligns with the sensor and temporal correlations results. However, if none of the existing segmentations can be identified, a new segmentation can be initialized with the incoming event. The objective of this step is to determine the beginning and the end of each activity. In this work, each beginning and end of each segment are identified when the sensors are highly correlated (i.e., sensor correlation is equal to 1) and when the difference between the start and end date of the activity does not exceed the maximum duration of each activity.Step 2: Identification of the activity’s triggering sensorOnce the beginning and the end of each activity are determined, the knowledge of the triggering sensor allows the method to recognize the activity. For example, for the sleeping activity, the beginning of the segment is at t = 00 H: 03: Min: 50 s and the end of the segment is at t = 05 H: 40: Min: 25 s. Figure 12 shows that during the test period, the sensor M003 is always the trigger sensor for the “sleeping” activity. On 4 November 2010, at t = 00 H: 03: Min: 50 s (i.e., the beginning of the segment), the trigger sensor is also M003, which belongs to the set of trigger sensors. Therefore, it is indeed the sleeping activity

Table 6 and Table 7 summarize the performance of the algorithm. The results show that the algorithm could segment several activities (sleeping, cleaning, cooking, etc.) with an F1 score of 0.63 to 0.99. Let us take the example of the “relax” activity. The F-score decreases between 4 November and 5 November 2010. The value goes from 0.830 to 0.75. We conclude that on an activity that has few events, and over very long durations, a badly segmented event will decrease the F-score.

## 7. Conclusions and Future Works

The recognition of human activities in a building is a problem of pattern recognition in irregular sampling of a time series. Different sensor event segmentation approaches in a smart building context are highlighted in the literature. These techniques provide sensor data representation for HAR algorithms. A time window method is commonly used, but in a smart home context, sensor data are often generated periodically over time. Consequently, a time window with fixed size is not feasible. The main challenge regarding dynamic segmentation based on data streaming is how to correctly identify whether or not two sensor events with a certain temporal separation belong to the same activity.

This paper presents a real-time event segmentation methodology for real-time human activity recognition in a building context. It is a two-step methodology: sensor and temporal correlation calculation, and the triggering sensor identification for each activity. The dynamic sensor event segmentation approach is performed by calculating the Pearson product–moment correlation (PMC) coefficient between events. The triggering sensor of the event is the very first sensor of an activity. In this work, it is not required to be discriminating (it is sufficient that it is the first), but it happens that we find that it is generally quite good at discriminating.

The real-time dynamic segmentation allows the method to determine the beginning and the end of each segment. Once they are determined, the knowledge of the trigger sensor allows the method to recognize the activity.

The methodology was validated on a real dataset. It has been demonstrated that the adoption of a trigger sensor identification and real-time dynamic windowing events can significantly achieve activity recognition with an F1-score of 0.63–0.99.

Future works are suggested as follows:In this work, for the sake of simplicity, the trigger sensor is considered as the first sensor that triggers the activity. It is used as the only discriminating sensor (i.e., this event makes it possible to clearly differentiate one activity from another). In this work, it is not required to be discriminating (it is sufficient that it is the first), but it happens that we find that it is generally quite good at discriminating. Future works will focus on the identification of trigger sensors using learning algorithms.The approach only considered several activities for a single person. It will be interesting to test the performance of this approach for concurrent activities where occupants’ activity profiles and their behaviors of using appliances change over time, which usually happens when the season or the calendar of an individual changes.The appliances, which are not only associated to occupants’ activities, are not addressed in this work (i.e., fridge, artificial lighting, etc.). It will be interesting to test the algorithm’s performance with activities that are related to electrical appliances.Occupants’ activities can affect many factors in houses, such as the electricity consumption, HVAC, and lighting systems. It will also be interesting to focus on the impacts on electricity consumption of domestic appliances.

Future research directions will be around the following:Testing this method on different hardware architectures, taking into account the cost of solutions.In a “digital twin” approach, a simulator will be used to overcome the difficulty of obtaining real data, and to generate synthetic data to introduce different variations (user habits, different sensors, different environment) according to a use case .

## Figures and Tables

**Figure 1 sensors-22-05458-f001:**
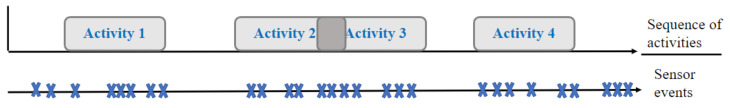
Sequence of activities.

**Figure 2 sensors-22-05458-f002:**
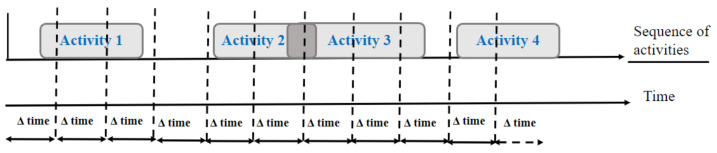
Time window.

**Figure 3 sensors-22-05458-f003:**
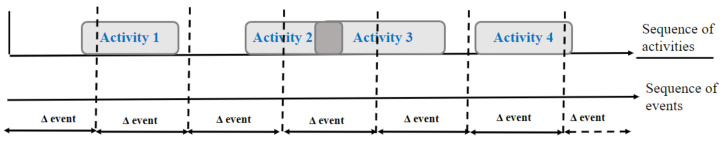
Sensor event window.

**Figure 4 sensors-22-05458-f004:**
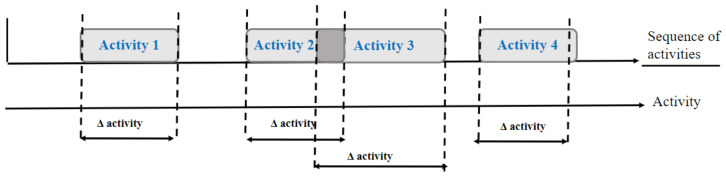
Drawing of the activity’s boundaries.

**Figure 5 sensors-22-05458-f005:**
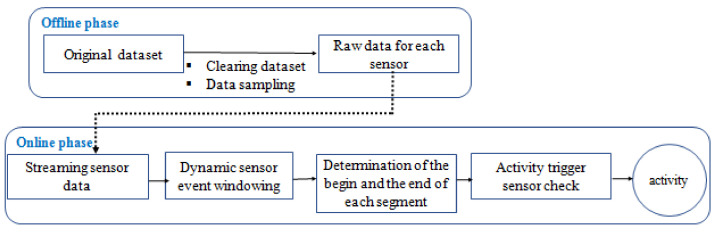
Diagram of the online HAR framework.

**Figure 6 sensors-22-05458-f006:**
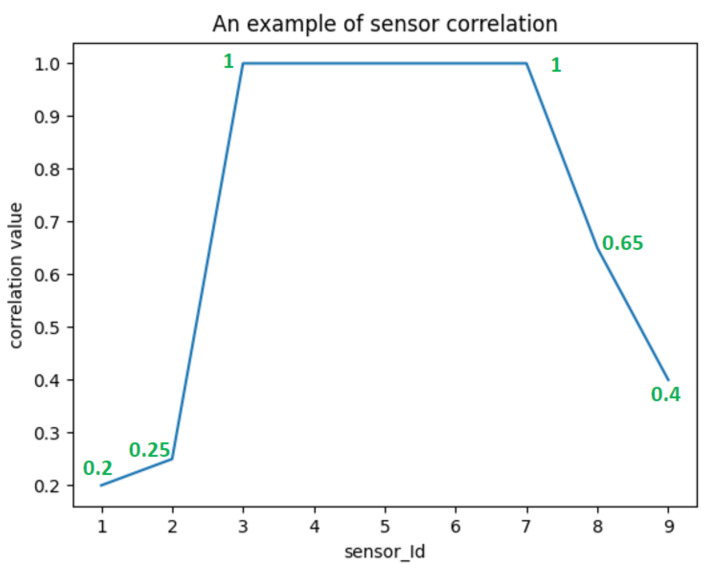
An example of sensor correlation.

**Figure 7 sensors-22-05458-f007:**
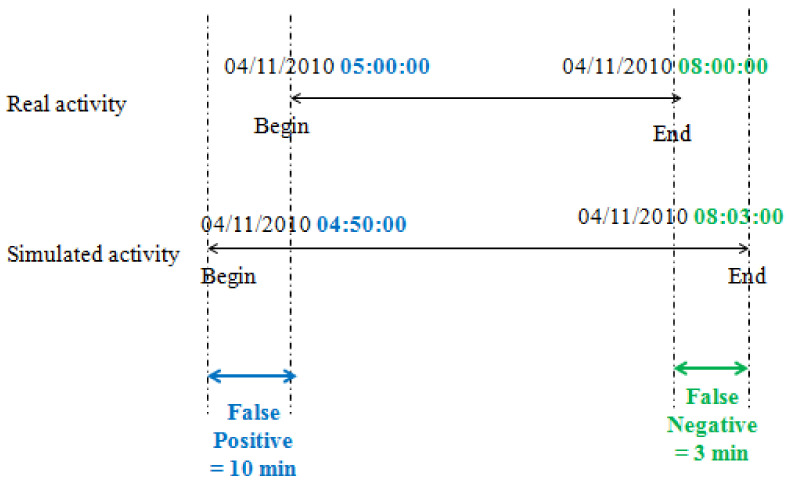
Precision and recall.

**Figure 8 sensors-22-05458-f008:**
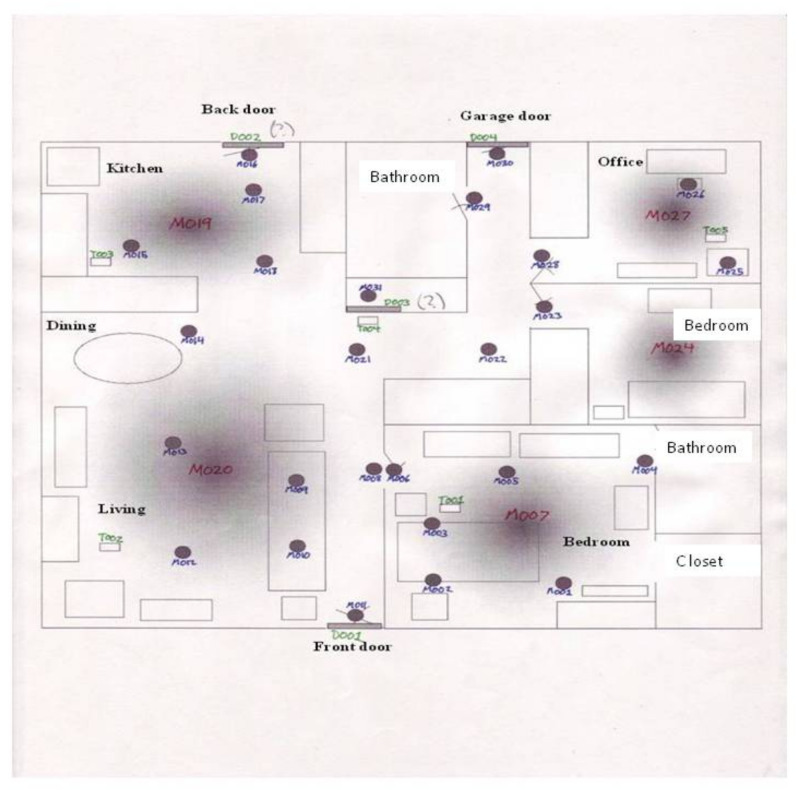
Sensor configuration of the dataset Aruba.

**Figure 9 sensors-22-05458-f009:**
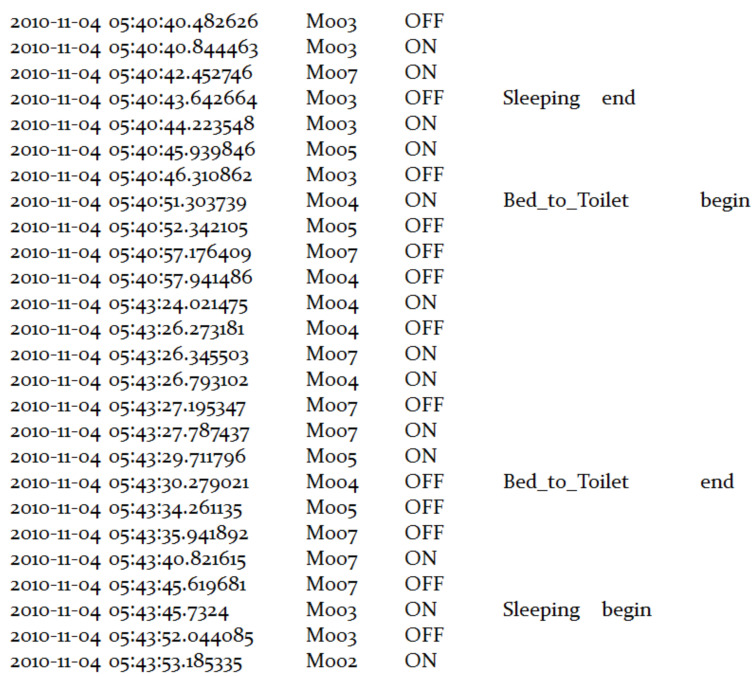
An extract of Aruba dataset with raw data.

**Figure 10 sensors-22-05458-f010:**
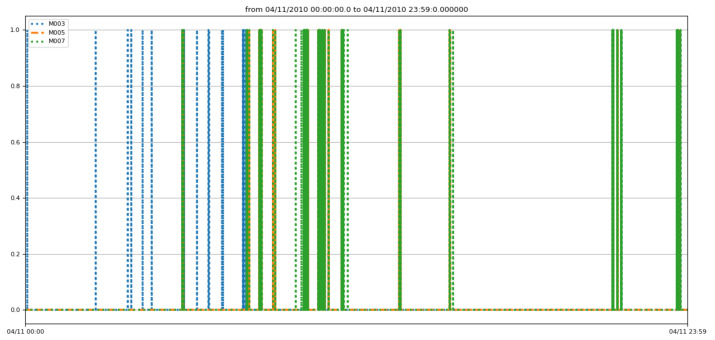
Sampled detection motions.

**Figure 11 sensors-22-05458-f011:**
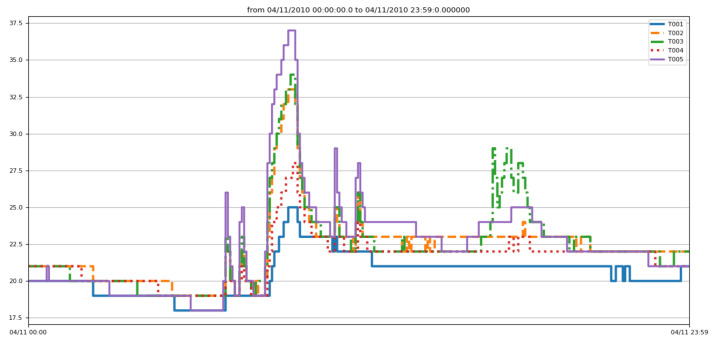
Sampling of temperature sensor events.

**Figure 12 sensors-22-05458-f012:**
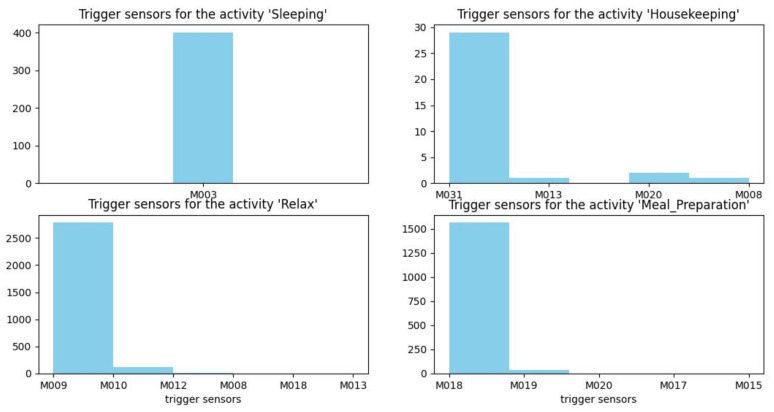
Trigger sensors for the activities sleeping, housekeeping, relaxing, and meal preparation.

**Figure 13 sensors-22-05458-f013:**
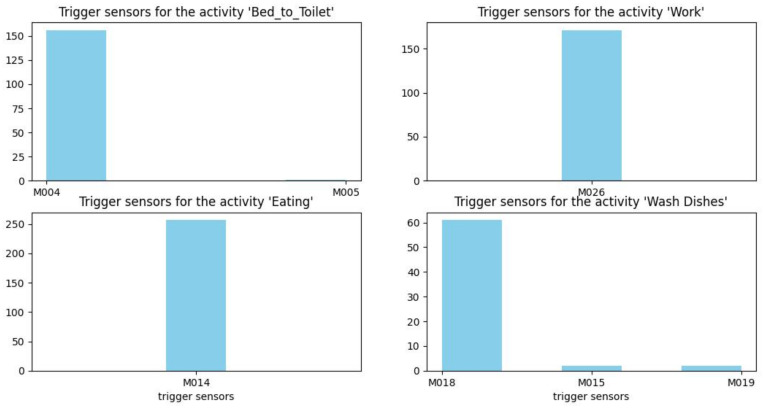
Trigger sensors for the activities bed to toilet, work, eating, and wash dishes.

**Figure 14 sensors-22-05458-f014:**
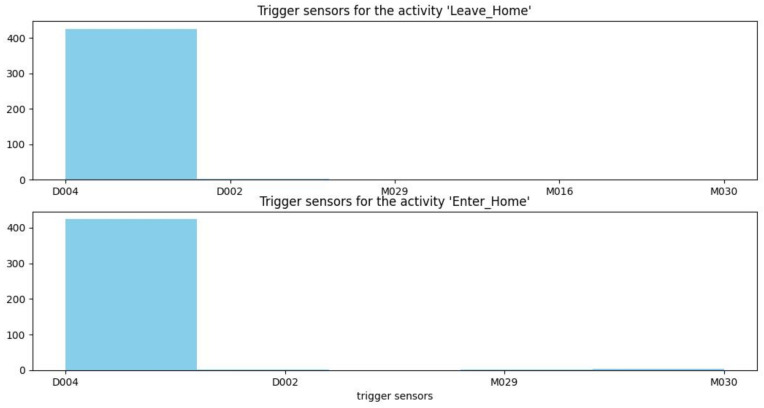
Trigger sensors for the activities leave home and enter home.

**Figure 15 sensors-22-05458-f015:**
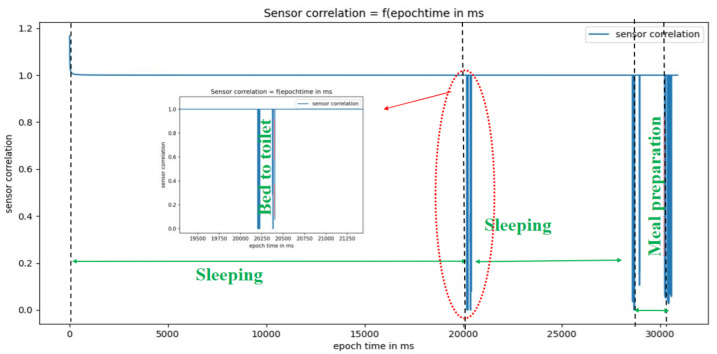
Segmentation of the activities sleeping, bed to toilet, and meal preparation.

**Figure 16 sensors-22-05458-f016:**
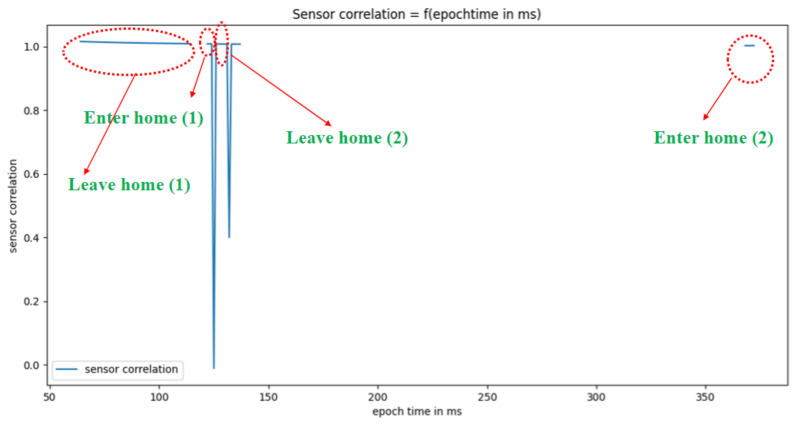
Segmentation of the activities leave home and enter home.

**Table 1 sensors-22-05458-t001:** Average duration of activities.

Activity	Average Duration
Sleeping	3:56:57
Bed to toilet	0:02:43
Housekeeping	0:20:10
Eating	0:09:55
Meal preparation	0:07:33
Work	0:17:04
Relax	0:33:23
Enter home	0:00:06
Leave home	0:00:06

**Table 2 sensors-22-05458-t002:** Timestamp comparison between real and simulated activities on 4 November 2010.

	Real Activity	Simulated Activity	Trigger Sensor
Sleeping (1)	Begin: 00 H: 03 MIN: 50 S	Begin: 00 H: 03 Min: 50 S	M003
	End: 05 H:40 MIN: 43 S	End: 05 H:40 MIN:25 S	
Sleeping (2)	Begin: 05 H:43 MIN:45 S	Begin: 05:44:03	M003
	End: 08 H: 01 MIN: 12 S	End: 08 H: 00 MIN: 23 S	

**Table 3 sensors-22-05458-t003:** F-score comparison between real and simulated activities on 4 November 2010.

	Accuracy	Recall	F-Score	Absolute Error
Sleeping (1)	1	0.99	0.99	9 s
Sleeping (2)	0.99	0.99	0.99	78 s
Average error	0.995	0.99	0.99	43.5 s

**Table 4 sensors-22-05458-t004:** Timestamp comparison between real and simulated activities on 4 November 2010.

	Real Activity	Simulated Activity	Trigger Sensor
Eating (1)	Begin: 09 H: 56 MIN: 41 S	Begin: 09 H: 56 MIN: 41 S	M014
	End:09 H: 59 MIN: 04 S	End: 09 H: 59 MIN: 09 S	
Eating (2)	Begin: 09 H: 59 MIN: 47 S	Begin: 10 H: 00 MIN: 41 S	M014
	End: 10 H: 02 MIN : 48 S	End: 10 H: 01 MIN: 31 S	
Eating (3)	Begin: 15 H: 25 MIN: 35 S	Begin: 15 H: 25 MIN: 15 S	M014
	End: 15 H:28 MIN:42 S	End:15 H:28 MIN:37 S	
Eating (4)	Begin: 17 H:35 MIN:16 S	Begin:17 H:36 MIN:26 S	M014
	End: 17 H:37 MIN:11 S	End:17 H:37 MIN:04 S	

**Table 5 sensors-22-05458-t005:** F-score comparison between real and simulated eating activities on 4 November 2010.

	Accuracy	Recall	F-Score	Absolute Error
Eating (1)	1	0.974	0.986	2.5 s
Eating (2)	0.576	0.538	0.555	71.5 s
Eating (3)	0.906	0.974	0.938	12.5 s
Eating (4)	0.5u27	0.917	0.966	38.5 s
Average error	0.752	0.850	0.861	31.25 s

**Table 6 sensors-22-05458-t006:** Activities recognition performance evaluation on 4 November 2010.

Activities	Accuracy	Recall	F-Score	Average Error
Sleeping	0.995	0.99	0.99	43.5 s
Bed to toilet	1	0.933	0.965	4.5 s
Housekeeping	0.989	0.968	0.978	13.5 s
Relax	0.876	0.847	0.830	118.75
Eating	0.752	0.850	0.861	31.25 s
Work	0.652	0.637	0.644	280.75 s
Leave Home	0.625	0.69	0.63	1.25 s
Enter Home	0.152	0.2	0.16	12.75

**Table 7 sensors-22-05458-t007:** Activities recognition performance evaluation on 5 November 2010.

Activities	Accuracy	Recall	F-Score	Average Error
Sleeping	0.999	0.99	0.99	25.16 s
Bed to toilet	0.844	0.948	0.892	15.5 s
Housekeeping	0.998	0.988	0.992	53.5 s
Relax	0.769	0.825	0.75	101 s
Eating	0.684	0.927	0.787	48.5 s
Work	0.639	0.627	0.569	178.85 s
Leave Home	0.715	0.59	0.64	5 s
Enter Home	0.714	0.565	0.59	4.5 s

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
