# Peer review of "Dynamic Segmentation of Sensor Events for Real-Time Human Activity Recognition in a Smart Home Context"

_sensors, 2022, doi:10.3390/s22145458_

Round 1
Reviewer 1 Report
This paper presents a sensor-based and heuristic human activity recognition method that solves the problem of pattern recognition and therefore activity event classification in event time series of arbitrary time length. The main contribution of this work is that it can recognize simultaneous HAR events in real-time and with high accuracy. The combination of three levels of hierarchy such as temporal, sensor correlation, and sensor triggering is interesting and provides an additional level of discrimination between different activity labels. This paper can be considered as a systems paper, as it concatenates more or less standard ideas to solve a known difficult problem.
In terms of topic, the paper is well suited to the Journal. It reports on an interesting study that is valuable. In general, the overall language is good, and the paper is presented with clarity and is easy to follow.
The related work is sufficient and is not a sterile presentation of previous work but rather a comprehensive presentation of their advantages and disadvantages. Finally, the experiments show that the method improved subjective and objective performance significantly.
Things to be definitely improved:
1. The quality of the reported images is extremely low. The authors should increase the quality of all images. The authors could possibly use vector graphics instead of bitmap to improve the quality of images.
2. The algorithm in Figure 7, should be presented not in image form but rather using an algorithmic package.
3. Figure 8 does not contain any valuable information and it should be excluded from the manuscript.
Author Response
Houda NAJEH
Postdoctoral researcher
Lab-STICC, IMT Atlantique, Brest & DeltaDore company
e-mail: houda.najeh@imt-atlantique.fr
Response to reviewer
Dear Reviewer:
I am pleased to resubmit for publication the revised version of “Dynamic Segmentation of Sensor Events for Real-Time Human Activity Recognition in a Smart Home Context “
I appreciated the constructive criticisms of your review. I have addressed each of the concerns as outlined below.
Following the reviewer’s advice, I have rewritten parts of the paper to provide more
clarity (see specific concerns outlined below).
First reviewer comments:
Specific Concerns
Things to be definitely improved:
- The quality of the reported images is extremely low. The authors should increase the quality of all images. The authors could possibly use vector graphics instead of bitmap to improve the quality of images.
Done. I used “\includegraphics[width=1\linewidth]{figure_name.PNG}” instead of “\includegraphics[scale]{figure_name.PNG}” for all the figures in the paper.
- The algorithm in Figure 7, should be presented not in image form but rather using an algorithmic package.
I used an algorithmic package to explain the real-time in dynamic windowing
- Figure 8 does not contain any valuable information and it should be excluded from the manuscript.
I eliminated this figure from the manuscript

Reviewer 2 Report
This paper proposes a three-step algorithm for real-time human activity recognition. In general, the organization and workflow are good with problem statement, formulations, and analysis. Before the recommendation of acceptance, authors are suggested to revise the paper with the comments below.
Comment 1. Abstract:
(a) Authors may specify more activities as stated in “several activities (sleeping, cleaning, cooking, etc.)”.
(b) Highlight the percentage improvement by proposed work, compared with the existing works.
Comment 2. Keywords, include more terms to better reflect the scopes of the paper.
Comment 3. Section 1 Introduction:
(a) Elaborate “Feedback to residents themselves can also, to be considered as added value”.
(b) Include references for the contents related to the Delta Dore.
Comment 4. Section 2 Problem statement and research objective:
(a) Elaborate literature review with the summary of the methodology, results, and limitations of the existing works.
(b) Regarding the settings of the windows, do they align with the common settings of the existing works?
(c) Consider to reorganize the objectives and contributions separately.
Comment 5. Section 3 Real-time recognition of human Activities in Smart Homes. For NILM, authors are suggested to consider to cite the following work (title as follows).
Handling data heterogeneity in electricity load disaggregation via optimized complete ensemble empirical mode decomposition and wavelet packet transform
Comment 6. Section 4 Proposed methodology for human activity recognition:
(a) Elaborate Figure 5 in main text.
(b) Clarify the two-times manipulation in equations (2) and (3).
(c) Table 1, consider to remove the decimal parts.
(d) Figure 7, consider to present the contents as normal algorithm.
(e) Figure 8, consider to present the contents as normal table.
Comment 7. Section 5 Case study:
(a) Share the challenges in recognitions of some activities.
(b) Compare the results with some existing works.
Comment 8. Elaborate future research directions.
Round 2
Reviewer 2 Report
I have one minor follow-up comment:
Comment 7. Section 5 Case study:
(b) Compare the results with some existing works
Done. I invite you to see subsection 5.6 in section 5.
Follow-up comment: More existing works should be included in the comparison.
